# Effects of Dark Chocolate Intake on Brain Electrical Oscillations in Healthy People

**DOI:** 10.3390/foods7110187

**Published:** 2018-11-08

**Authors:** Efraín Santiago-Rodríguez, Brenda Estrada-Zaldívar, Elba Zaldívar-Uribe

**Affiliations:** 1Diagnóstico, Tratamiento e Investigación Neurológica, S. C. Querétaro, Querétaro 76177, México; ezu140267@gmail.com; 2Facultad de Ciencias Naturales, Universidad Autónoma de Querétaro, Querétaro 76000, México; e.zbrenda00@gmail.com

**Keywords:** dark chocolate, electroencephalogram, cognitive effects

## Abstract

Dark chocolate is rich in flavonoids that can have effects on body composition and cognitive performance. The aim of this study was to analyze the effects of acute and subchronic chocolate intake on electrical brain oscillations. A study with 20 healthy subjects (mean age of 24.15 years) and a control group with five subjects (mean age of 23.2 years) was carried out. In the acute effect study, the subjects’ intake was dark chocolate (103.72 mg/kg of body weight) rich in flavonoids and low in calories as in fasting. In the control group, the subjects intake was only low-calorie milk. For the subchronic effect, a daily dose of dark chocolate was given for eight days. The baseline electroencephalogram (EEG) was recorded before dark chocolate intake; at 30 min, the second EEG was carried out; on the eighth day, the third and fourth EEGs were performed before and after the last intake. In acute and subchronic intake, Delta Absolute Power (AP) decrease was observed in most brain regions (*p* < 0.05), except in the right fronto-centro-temporal regions. In the Theta band, there was a generalized decrease of the AP of predominance in the left fronto-centro-temporal regions. In contrast, an increase in AP was observed in the temporo-occipital regions in the Alpha band, and in the right temporal and parieto-occipital regions in the Beta band. The control group did not have significant changes in brain oscillations (*p* > 0.05). We concluded that acute and subchronic chocolate intake decreased the Delta and Theta AP and increased Alpha and Beta AP in most brain regions.

## 1. Introduction

In recent years, one of the major problems of global public health has been the growth of the population over 65 years old, and the increase in aging-related diseases such as mild cognitive impairment and dementias [1]. On the other hand, in the first years of life, increases in the availability of neonatal intensive care units have resulted in an increase in the number of surviving premature and very-low-birth-weight infants with perinatal brain injury (PBI) [2]. Therefore, one of the current scientific challenges in these two population groups is to modify aging and to prevent and eventually reverse cognitive impairment or enhance cognitive performance [3]. Thus, the chronic intake of flavonoids has been demonstrated to have beneficial effects for cardiovascular disease and many others, which include decreasing blood pressure in hypertensive patients, and decreasing insulin resistance and overweight in patients with metabolic syndrome. Chocolate, especially dark chocolate with its high content of flavonoids and theobromine, has proven to be a promising food for cardiovascular and metabolic effects [4]. 

Recently, some papers have shown that the flavonoids of dark chocolate improve cognitive functions, specifically working on memory and Trail-Making-Tests in healthy people and those with cognitive impairment [5,6]. Dark chocolate has the highest flavonoid content. Some commercial preparations have modified the composition of dark chocolate to increase the flavonoid content and decrease the caloric intake. Analysis of the effects of dark chocolate has been divided according to the days in which it is ingested in acute, subchronic and chronic conditions [7].

Although some effects of dark chocolate on cognitive performance have already been studied, research on the modifications of brain electrical activity after the acute intake of chocolate has been poorly analyzed. It is well established that by taking advantage of such powerful electroencephalographic (EEG) technology one could easily probe brain electrical oscillations. With the use of quantitative EEG (qEEG) for the analysis of brain electric activity, precision and consistency were improved [8,9,10]. Currently, to our knowledge, there are no papers that study the acute effect of dark chocolate on brain electric oscillations in healthy people. Therefore, the objective of our study was to analyze how brain oscillations are affected by the acute intake of chocolate rich in flavonoids and low in calories, as well as the intake of one cup of chocolate every morning for seven days. 

## 2. Materials and Methods

A prospective longitudinal study was conducted. Twenty healthy subjects (13 female and 7 male), volunteers, who attended the specialized center (Diagnóstico, Tratamiento e Investigación Neurológica, S.C.) were included in groups of five people every two weeks to complete the 20 subjects. The control group included 5 healthy subjects (3 female and 2 male). Subjects in the study group ingested low-calorie milk with dark chocolate, while the control group ingested only low-calorie milk. Characteristics of the study were fully explained to each participant, as well as the minimum risks of participating in the project; those who agreed signed an informed consent form, and underwent clinical nutritional and neurological examinations. The project followed the Helsinki guidelines and was approved by Ethical Research Committee of Diagnóstico, Tratamiento e Investigación Neurológica, S.C. (protocol # 2016/001; date of approval: 28 November 2016). 

### 2.1. Baseline EEG Recording 

The baseline EEG was recorded over twenty minutes; patients closed their eyes but stayed awake during the procedure, and the data were collected using the Medicid-5 system (Neuronic Mexicana, SA, México City, México). Amplifier characteristics were as follows: 10,000 dB gain, low-cut filters at 0.05 Hz and high-cut filters at 70 Hz. Nineteen referential leads of the International 10/20 System with linked earlobes as references were used (Figure 1). The impedance was under 5 Kohms in all electrodes with a sampling frequency of 200 Hz. All EEG recordings were analyzed by clinic neurophysiologists (ES-R). The analysis of background EEG activity was carried out with the Fast Fourier Transform. Twenty-four segments of 2.56 s each were selected in blind form by the principal author (ES-R). Only the segments in which the patient was totally awake were analyzed to obtain the absolute power (AP), relative power (RP) and mean frequency (MF) of Delta (0.5–3.5 Hz), Theta (3.6–7.5 Hz), Alpha (7.6–12.5 Hz) and Beta (12.6–19 Hz) bands. The subtraction of the global scale factor (GSF) was applied to AP in all bands to decrease non-physiological variability; this procedure is known to improve diagnostic precision [11]. 

### 2.2. Experimental Maneuver 

The experimental maneuver consisted of ingesting 6.8 g of dark chocolate rich in flavonoids (375 mg) and low in calories (25 kcal) CocoaVia (Mars Symbioscience Inc., Germantown, MD, USA) dissolved in 250 mL of low-calorie milk (115 kcal, Milk and Derivatives Trader, SA CV, Torreón, México). Subjects in the control group only ingested 250 mL of low-calorie milk. After thirty minutes of chocolate intake or low-calorie milk, the second EEG was performed with the same characteristics as the baseline EEG recording. For the subchronic effect, a daily dose of dark chocolate was given for eight days, and the third EEG was performed after the last intake on day eight.

### 2.3. Statistical Mnalysis 

Statistical analysis was performed using descriptive measures, mean and standard deviation. For the analysis of brain electrical activity, the baseline values of AP and MF of Delta, Theta, Alpha and Beta bands were normalized to 100%, and the percentage of variation after dark chocolate intake was comparable among all subjects. Both crude and normalized values were compared by means of a one-way repeated measures ANOVA, and two conditions (before and after dark chocolate intake) were used for analyses in Neuronic Statistics (Neuronic Mexicana, SA, México City, México, 2011), a specialized statistical program for EEG analysis. Values of *p* < 0.05 were considered significant.

## 3. Results

In the chocolate group we studied 20 healthy subjects; 7 males and 13 females, and the subjects had a mean age of 24.15 ± 4.84 years. Subjects were free of metabolic and nervous system diseases, although two of them were overweight (body mass index of 32.1 and 35.4). In the control group, 5 healthy subjects with a mean age of 23.2 ± 2.58 years were studied (3 female and 2 male). The results of the effect of chocolate intake or low-calorie milk on brain electrical activity are described in the next paragraphs. 

### 3.1. Acute Effect of Dark Chocolate Intake

Modifications of brain electrical activity after intake of 250 mL (103.72 mg/kg of body weight) of dark chocolate were observed in AP and MF in various leads and in different frequency bands (Table 1). Delta AP decrease was observed in most brain regions (*p* < 0.05, *n* = 20, ANOVA-test), except in the right fronto-centro-temporal regions, where there were no significant changes. Additionally, in the Theta band, there was a generalized decrease in the AP of predominance in the left fronto-centro-temporal regions. In contrast, in the Alpha and Beta bands, an increase in AP was observed. In the Alpha band, this increase was localized in the temporo-occipital regions, and in the Beta band, this increase was localized in the right temporal and parieto-occipital regions in bilateral form (Figure 2). The most significant values were observed in the occipital leads, and the Alpha and Beta bands are described in detail in the following paragraphs. The baseline Alpha AP of the O1-AVR lead was 49.68 ± 51.71 μV2/Hz after the chocolate intake was increased by 32.69% (*p* = 0.04). The Alpha AP of the O2-AVR lead was 62.12 μV2/Hz; after ingesting the chocolate, an increase of 29.01% was observed (*p* = 0.009). The baseline Beta AP of the O1-AVR lead was 4.50 ± 1.85 μV2/Hz, and post-chocolate intake increased by 26.33% (*p* = 0.0001). In the O2-AVR lead, baseline was ±2.14 μV2/Hz, and a significant increase of 17.21% (*p* = 0.0001) was observed after chocolate intake (Figure 3 and Figure 4).

### 3.2. Acute Effect of Low-Calorie Milk Intake

After the ingestion of 250 mL of low-calorie milk, no significant differences were observed between the baseline values of the AP of the Delta, Theta, Alpha and Beta bands in any of the brain regions analyzed (*p* > 0.05, *n* = 5, ANOVA-test). In contrast to the chocolate group, in the control group the alpha and beta AP did not change significantly in the occipital regions. The baseline Alpha AP of the O1-AVR lead was 38.20 ± 8.49 μV2/Hz after the low-calorie milk intake was 39.40 ± 13.25 μV2/Hz (*p* = 0.73). The Alpha AP of the O2-AVR lead was 48.68 μV2/Hz; after ingesting the low-calorie milk was 42.84 ± 23.00 μV2/Hz (*p* = 0.18). The baseline Beta AP of the O1-AVR lead was 4.72 ± 2.73 μV2/Hz; after ingesting the low-calorie milk was 4.39 ± 2.03 μV2/Hz (*p* = 0.40). In the O2-AVR lead, baseline was 4.24 ± 1.78 μV2/Hz, and a non-significant decrease to 3.81 ± 1.28 μV2/Hz (*p* = 0.16) was observed after low-calorie milk intake (Figure 3 and Figure 4).

### 3.3. Subchronic Effect of Seven-Day Dark Chocolate Intake

To evaluate the subchronic effect of daily chocolate intake, the baseline AP value in the O1 and O2 leads was compared with those obtained seven days later. In the O1-AVR lead, the Alpha AP increased to 63.39 ± 58.23 μV2/Hz, which represented 30.51% (*p* = 0.007). The AP Alpha of the O2-AVR lead increased to 72.85 + 65.13 μV2/Hz with an increase of 49.52% (*p* = 0.023). Finally, the Beta AP of the O1-AVR lead increased to 5.16 ± 1.69 μV2/Hz, representing 19.0.2% (*p* = 0.005). In the O2-AVR lead, the increase was 5.65 ± 2.87 μV2/Hz, representing 28.09% (*p* = 0.03) (Figure 3 and Figure 4). 

### 3.4. Acute Effect on a Subchronic Effect of Dark Chocolate Intake

The Alpha AP of the O1-AVR lead increased to 71.23 ± 61.36 μV2/Hz, corresponding to 54.01% (*p* = 0.0003). In the O2-AVR lead, the AP increased to 78.12 ± 61.51 μV2/Hz, corresponding to 71.88% (*p* = 0.007). Finally, the Beta AP of the O1-AVR increased to 6.12 ± 2.13 μV2/Hz, corresponding to 40% (*p* = 0.000009). In the O2-AVR lead, the AP increased to 6.62 + 3.55 μV2/Hz, corresponding to 47.28% (*p* = 0.004) (Figure 3 and Figure 4). On day eight, after the last chocolate intake, the increase in Alpha and Beta AP was higher than basal AP and the AP after seven days of intake.

### 3.5. Modifications of Mean Frequency

After dark chocolate intake, no modifications in MF Delta, Theta, and Beta were observed. Only a generalized decrease of MF Alpha was observed, with more significant values in occipital regions (Figure 2). In spectral analysis of EEG, the characteristic Alpha spectral peak was moved to lower frequencies in the majority of subjects, and this modification was more evident at seven days, and after the last dark chocolate intake at day eight. In less consistent form, an increase in the amplitude of the spectral Alpha peak was found. 

## 4. Discussion

The main results of our study were that intake of 6.8 g of dark chocolate is enough to cause a decrease in the AP Delta and Theta, and an increase in the AP Alpha and Beta in the temporo-occipital leads without changes of brain oscillations in the control group. In addition, a decrease in MF Alpha was found. These results will be analyzed in the following paragraphs.

Chocolate, especially dark chocolate, with its high content of flavonoids, has proven to be a promising food in terms of its cardiovascular effects. Recently, some studies have reported cognitive effects after intake of dark chocolate, specifically on attention and working memory. To date, only one study has reported the effects of dark chocolate on brain electrical activity, specifically stable-state visual evoked potentials [12]. Moreover, no study has reported the effects of chocolate on human brain electrical oscillations recorded in the EEG. In our study, we analyzed the acute effect of the intake of a single dose of dark chocolate and the subchronic effect of daily intake for seven and eight days. 

In the acute effect analysis, there were notable changes in brain electrical activity 30 min after dark chocolate intake without changes of brain oscillations in the control group. It has been postulated that the cognitive acute effects appear a few minutes after dark chocolate intake and can be mediated indirectly by vascular mechanisms that cause an increase in cerebral blood flow. In recent studies, it was shown that dark chocolate intake with a mean dose of 516 mg and a high dose of 903 mg of flavonoids increases brain blood flow as evaluated by functional magnetic resonance [13] and near-infrared spectroscopy [14]. However, these studies show conflicting results if the increase in cerebral blood flow is mediated by nitric oxide-induced vasodilation or not. Regarding the cognitive effects of chocolate intake, a double-blind crossover study of three groups was conducted: the first with high-dose flavonoids (994 mg), the second with 520 mg, and a control group with 46 mg. A working memory task of serial digital subtraction before and after the intake of chocolate was applied. Subjects with medium and high doses of flavonoids showed significant improvement in this work memory task in relation to the control group [15]. Other studies have had similar results [16], and negative results [17]. On the other hand, the effects found with dark chocolate could be attributed to the low-calorie milk in which the chocolate was dissolved. However, in the control group where the subjects ingested the same amount of low-calories milk but without dark chocolate, no significant changes in the brain oscillations were observed.

On the other hand, the acute effect of dark chocolate intake seems to have a direct relationship with the demand for cognitive resources of the task or with the interference or damage of the neural circuits involved in the task. This was observed in the study by Grassi et al. [18], in which the dark chocolate effect was compared in a group of subjects with sleep deprivation and a control group with normal sleep submitted to a working memory 2-back test. The results showed that the intake of a single dose of chocolate reversed the deleterious effects of sleep deprivation and even improved performance of tasks on the basal level. Consistent with previous results, in our study, the highest increases in Alpha and Beta AP were observed in those subjects with the lowest values at baseline. Therefore, in future studies, it would be desirable to evaluate whether similar doses of dark chocolate flavonoids produce different effects in populations of healthy subjects, with cognitive impairment and learning or neurodevelopmental disorders. 

In addition to vascular changes and cerebral blood flow, it is possible that dark chocolate effects occur through neural networks involved in the processes of attention, work memory or episodic memory, and that these effects can be acute or subchronic. Flavonoids cross effectively the blood brain barrier, and many studies have found them in brain regions that are actively involved in the processes of attention, episodic, procedural and working memory, such as the hippocampus, frontal cortex and basal ganglia [19]. In an experimental study, the authors analyzed whether epicatechin, a dark chocolate flavonoid, could enhance memory formation of Lymnaea snail exposed to epicatechin before training, during the consolidation period immediately following training and 1 hour after training. Only the snails exposed to epicatechin immediately after training showed enhanced memory formation [20]. In addition, in a mouse experimental model, the effects of the chronic consumption of food enriched in cocoa-based dark chocolate on the excitability of hippocampal neuronal networks were analyzed. A significant enhancement in seizure-like population spike bursting at the neurogenic dentate gyrus was found with a reduction in the levels of GABA-α1 receptors [21]. 

It is necessary to emphasize that in our study, although the effects of dark chocolate on electrical brain activity were recorded in almost all brain regions, they were more evident in the temporo-occipital regions for the Alpha band, and in the parieto-occipital and temporal regions in the Beta band. In these regions, the neural circuits are actively involved in visuospatial working memory and episodic memory processes. In addition, the Alpha frequencies have been implicated in visuospatial working memory processes especially in the maintenance phase, where it has been observed that the higher the complexity of the task and the greater the demand for cognitive resources, the Alpha spectral peak has a greater amplitude, indicating that a greater number of neurons are activated synchronously [22,23,24].

Our study has some limitations; for example, the sample size was small, especially in the control group; however, it was sufficient to detect changes in brain electrical oscillations in chocolate-ingesting groups. 

## 5. Conclusions

We can conclude that the results of our study sustain that acute and subacute dark chocolate intake decreases the AP of the Delta and Theta bands, and increases the AP of the Alpha and Beta bands in most brain regions, especially in the temporo-occipital leads. The control group, with low-calorie milk intake, did not have significant changes in brain oscillations.

## Figures and Tables

**Figure 1 foods-07-00187-f001:**
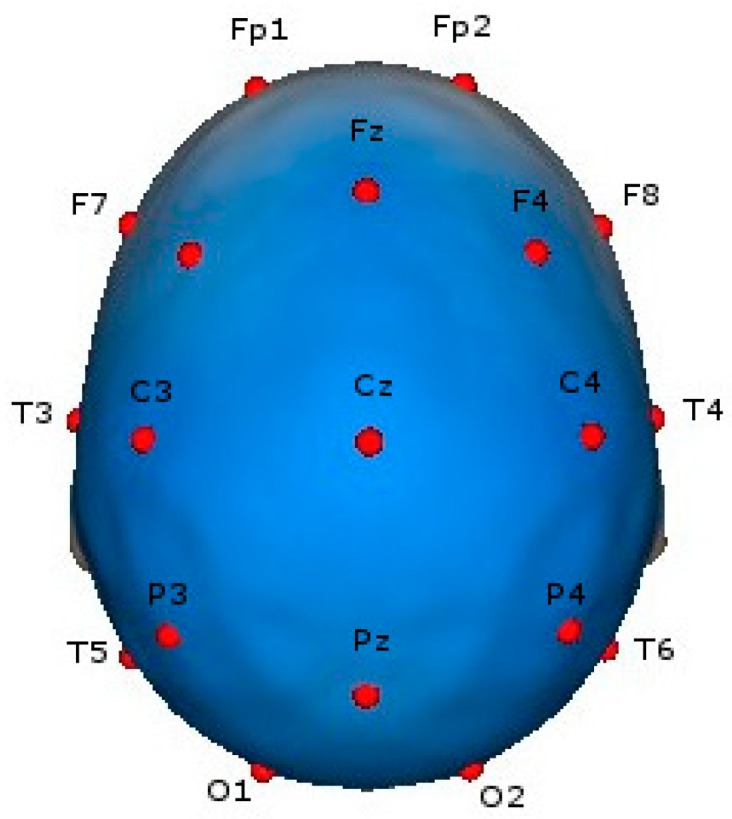
Position of 19 electroencephalogram (EEG) electrodes with referential leads of the International 10/20 System with linked earlobes as references.

**Figure 2 foods-07-00187-f002:**
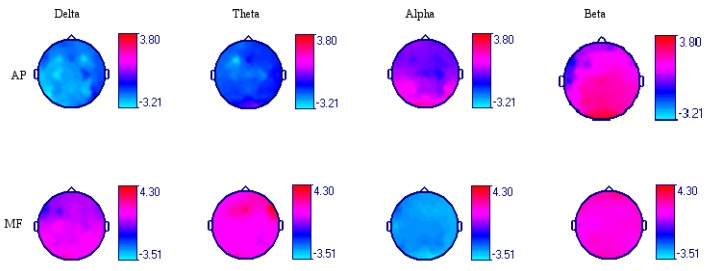
Comparison of the Absolute Power (AP) and the Mean Frequency (MF) of the EEG frequency bands before and after the acute dark chocolate intake. The *T* values are observed with a significance threshold of −1.73 to 1.73 (*p* = 0.05, *n* = 20, ANOVA-test).

**Figure 3 foods-07-00187-f003:**
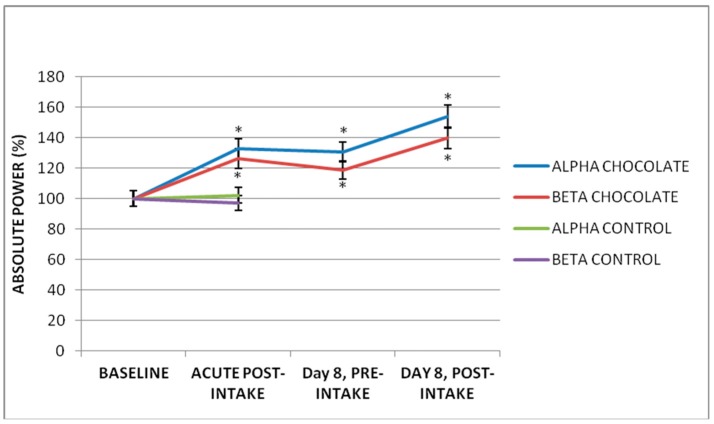
Effects of dark chocolate intake on the AP of the EEG bands in the O1-AVR lead. A significant increase in the Alpha and Beta bands was observed on the first day, after its daily intake for seven days and finally after the last intake at day eight (* *p* < 0.05, *n* = 20, ANOVA-test). In the control group, no significant changes were observed (*p* > 0.05, *n* = 5, *T*-test).

**Figure 4 foods-07-00187-f004:**
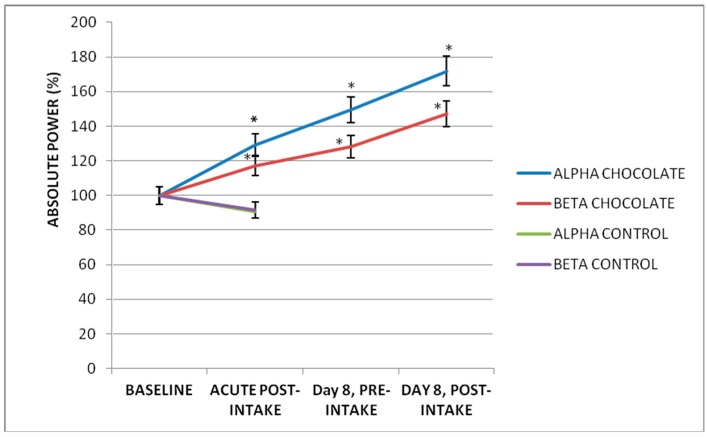
Effects of dark chocolate intake on the EEG bands in the O2-AVR lead. An increase of the Alpha and Beta AP was observed after the ingestion of a dark chocolate, after its daily intake for seven days and finally after the last intake of it at day eight. The Alpha band showed a higher percentage of change after the dark chocolate intake (* *p* < 0.05, *n* = 20, ANOVA-test). In the control group, no significant changes were observed (*p* > 0.05, *n* = 5, *T*-test).

**Table 1 foods-07-00187-t001:** Acute effect of dark chocolate intake, *T* values obtained by comparing the basal values of the Delta, Theta, Alpha and Beta Absolute Power (AP) against the values obtained after the dark chocolate intake. *T* threshold of −1.73 to 1.73 (*p* = 0.05, *n* = 20, ANOVA-test).

Leads	AP Delta	AP Theta	AP Alpha	AP Beta
Fp1-AVR	−1.566	−1.906 *	0.332	0.352
Fp2-AVR	−2.073 *	−1.409	−0.122	0.042
F3-AVR	−2.795 *	−3.035 *	−0.349	0.128
F4-AVR	−2.905 *	−1.323	−0.521	1.681
C3-AVR	−2.994 *	−2.108 *	0.295	2.027 *
C4-AVR	−1.378	−2.196 *	−0.791	2.110 *
P3-AVR	−2.172 *	−1.431	1.407	2.494 *
P4-AVR	−2.776 *	−1.204	0.058	2.191 *
O1-AVR	−2.680 *	−0.395	2.557 *	3.356 *
O2-AVR	−1.796 *	0.318	2.061 *	3.800 *
F7-AVR	−1.499	−1.809 *	−0.168	−1.000
F8-AVR	−1.559	−1.273	−0.177	0.66
T3-AVR	−2.817 *	−1.967 *	0.443	−0.915
T4-AVR	−1.509	−0.397	0.076	1.741 *
T5-AVR	−3.213 *	−1.541	1.768 *	1.381
T6-AVR	−0.878	−1.222	2.134 *	2.088 *
FZ-AVR	−1.577	−0.878	0.000	1.663
CZ-AVR	−2.354 *	−2.154 *	−0.595	2.528 *
PZ-AVR	−2.607 *	−1.851 *	0.312	2.493 *

Note: * Significant values with *p* < 0.05.

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
