# Peer review of "Effects of Dark Chocolate Intake on Brain Electrical Oscillations in Healthy People"

_foods, 2018, doi:10.3390/foods7110187_

Round 1
Reviewer 1 Report
The article "Effects of dark chocolate intake on brain electrical oscillations in healthy people" is a very interesting article in which authors found that both acute and sub-chronic dark chocolate intake increased Alpha and Beta absolute power (AP) in most brain regions while recording in EEG, while it reduced delta and Theta AP. However, there are few weakness or issues in the article that need to be addressed as discussed below.
Line 13-14: "An open study with 20 healthy subjects was carried out". This sentence is too short, you may expand it by introducing both genders were included in the study or specify the age range, range for details. You may also mention if any subject were fasting or not.
Line 15: It will be more clear if you could provide the actual dose of substance/body weight administered.
Line 15-18: There is a room to re-arrange this line 15-18 considering consistency in writing style for good flow of text.
Line 18: since p- value has been provided, it will be better to define n and statistical test used in parenthes. If you consider upper case N= subject, perhaps lower case 'n' = EEG trace. Here, I assume it could be average EEG traces provided.
Line18-22: It would be also more clear if you could make connection of result corresponding to the dark chocolate intake with respect to time frame. For example, at which time frame this data was obtained is not explicitly clear here in result section.
Line 40: I do not think it is "tail-making tests" but I guess it should be Trail-Making Tests (TMT).
Line 47-48: Please remove "In this way" from this sentence. I would rather reorganized this sentence by mentioning - It is well established that taking an advantage of such powerful EEG technology, one could easily probe brain electrical oscillations or similar type of texts.In results section, it would be better provide one schematic showing where the electrode position.
Line 55-61: You may specify few things such as gender of subject, the consent from the subject for this publication, institution approval for this study.
Line 71-74: How would you confirm that subject were fully awake?
Did you ask questionnaire at the end of recording session to make sure they did not sleep during 20 min? The characteristics slow waves in EEG signal further confirms whether they are not slept. If pupil, are closed but mental activity are still working, then you may discuss whether cholinergic signalling or adrenergic signalling could highlighted in your discussion section.
You may also provide any data in tabular form or pie chart to illustrate if in case any subject felt slept out of total.
Line 81: How would you overrule that low-calorie milk did not influenced your results? Drinking liquids signals might activate different specific brain centers.
Line 96: replace "seven" with "7" to make consistency.
Line 98: Can you provide mean weight of subject and two overweight mean value?
Line 104: define test in parenthes after p value.
Line 144-145: Text in the Figure 1 in the figure itself is slightly blur.
Line 148-151: Figure 2 and Figure 3: Please insert "*" on the graphs wherever the data are significant. The text in x-axis and y-axis are so faint. You may want to bold it and increase the font so that we can read it.
Whenever possible these parameters like p, n, and test used should come together.
Line 156: 'one dose' may be replaced with exact quantity.
Line 157: Check "fund" are appropriate word.
Line 185: The wording 'n 2-back' is harder for me to make sense. Please check.
Line 198: check wording "exoposed it".
Line 220: Provide subheading "Conclusion". State your overall result making consistent with how you have defined in 'Abstract' Section of the manuscript.
Line 231-289: Make sure that references follows the journal guidelines. There looks inconsistency currently.
Beside above other issues are:
If it is absolute power, what does negative value means, please mention it in your results. You may also discuss the relative power of EEG signal with respect electrode position.
Control is very important for this type of experiment. Generally (n=3 or 4) will also be acceptable for this purpose, please provide it.
Author Response
October 29th, 2018
Ms. Christine Fu
Assistant Editor, Foods
Dear Ms Christine Fu:
In response to the comments of the reviewers regarding our manuscript ID: Foods-380411 entitled “Effects of dark chocolate intake on brain electrical oscillations in healthy people”, we have made the following corrections to the manuscript (see below). We have included the revised version of the manuscript.
Reviewer 1
We appreciate your thorough review of our paper. We´ve done the corresponding changes and answered questions raised.
Line 13-14: "An open study with 20 healthy subjects was carried out". This sentence is too short, you may expand it by introducing both genders were included in the study or specify the age range. range for details. You may also mention if any subject were fasting or not.
The sentence was rewritten
Line 15 : It will be more clear if you could provide the actual dose of substance/body weight administered.
The actual dose of substance/body weight administered was added to manuscript.
Line 15-18: There is a room to re-arrange this line 15-18 considering consistency in writing style for good flow of text.
The sentence was rewritten
Line 18: since p- value has been provided, it will be better to define n and statistical test used in parenthes. If you consider upper case N= subject, perhaps lower case 'n' = EEG trace. Here, I assume it could be average EEG traces provided.
We agree with the observation, but with the corrections requested in the above lines we have exceeded the 200 words of the summary requested by the Journal. Therefore, we think is better to keep the statistical test and the N only in the Material and Methods section.
18-22 It would be also more clear if you could make connection of result corresponding to the dark chocolate intake with respect to time frame . For example, At which time frame this data was obtained is not explicitly clear here in result section.
The sentence was rewritten
Line 40: I do not think it is "tail-making tests" but I guess it should be Trail-Making Tests (TMT).
The mistake was corrected
Line 47-48: Please remove "In this way" from this sentence. I would rather reorganized this sentence by mentioning - It is well established that taking an advantage of such powerful EEG technology, one could easily probe brain electrical oscillations or similar type of texts.In results section, it would be better provide one schematic showing where the electrode position.
The suggested text was added to the introduction and the figure of the position of the electrodes was added to the Material and Methods section.
Line 55-61: You may specify few things such as gender of subject, the consent from the subject for this publication, institution approval for this study.
The gender of the subjects was added. The informed consent and institutional approval were already described in the text.
71-74: How would you confirm that subject were fully awake?
As described in the Material and Methods section, only segments where the patient was fully awake were used for quantitative analysis. The above is achieved by the careful selection of EEG segments where alpha activity is undoubtedly present. Doubtful or transitional segments to the N1 phase of sleep (drowsiness) with decreased alpha activity and increased theta activity are discarded. This precaution is taken in most studies of quantitative EEG analysis
Did you ask questionnaire at the end of recording session to make sure they did not sleep during 20 min? The characteristics slow waves in EEG signal further confirms whether they are not slept. If pupil, are closed but mental activity are still working, then you may discuss whether cholinergic signalling or adrenergic signalling could highlighted in your discussion section.
No questionnaire was applied to determine if the patients had slept. It is known from polysomnographic studies that the opinion of the patient regarding the quality and quantity of sleep is imprecise and subjective. Therefore, we prefer to use the previously described technique that has proven to be more sensitive, accurate and widely used in studies of cognitive processes with EEG. On the other hand, we agree that it is interesting to analyze the effect of dark chocolate on autonomic, sympathetic and parasympathetic activity and pupillary diameter variations. However, we believe that this problem, although interesting, can be confirmed in future studies.
You may also provide any data in tabular form or pie chart to illustrate if in case any subject felt slept out of total.
It is not feasible for the reasons explained in the previous answers.
Line 81: How would you overrule that low-calorie milk did not influenced your results? Drinking liquids signals might activate different specific brain centers.
We agree that the open design of the study does not eliminate the possibility that milk low in calories is responsible for the effect obtained. To resolve this possibility, we accepted the suggestion of the Reviewer and carried out a small control group with 5 healthy subjects who ingested 250 ml of low-calorie milk without dark chocolate. Due to problems of feasibility of time and resources we only carried out the study of acute intake. As expected, there were no significant differences in brain electrical activity before and after the ingestion of 250 ml of low-calorie milk. The data from the analysis of the control group were added in the Material and Methods, Results and Discussion sections as well as in Figures 2 and 3.
Line 96: replace "seven" with "7" to make consistency.
The seven was replaced by 7
Line 98: Can you provide mean weight of subject and two overweight mean value?
The body mass index of the 2 patients were added.
Line 104: define test in parenthes after p value.
The statistical test was added
Line 144-145: Text in the Figure 1 in the figure itself is slightly blur.
The quality of Figure 1 is the best provided by the software for the statistical analysis of the EEG
Line 148-151: Figure 2 and Figure 3: Please insert "*" on the graphs wherever the data are significant. The text in x-axis and y-axis are so faint. You may want to bold it and increase the font so that we can read it.
In Figures 2 and 3 asterisks were inserted to indicate the significant values. We improve the quality of the labels of the x, y axes with bold text.
Whenever possible these parameters like p, n, and test used should come together.
The suggestion was carried out.
Line 156: 'one dose' may be replaced with exact quantity.
The suggestion was carried out.
Line 157: Check "fund" are appropriate word.
The mistake was corrected
Line 185: The wording 'n 2-back' is harder for me to make sense. Please check.
The mistake was corrected. The word “test” was added. The correct form is “working memory 2-back test”
Line 198: check wording "exoposed it"
The mistake was corrected
Line 220: Provide subheading " Conclusion". State your overall result making consistent with how you have defined in 'Abstract' Section of the manuscript.
The “Conclusion” subheading was added and modified to be consistent with the “Abstract”.
Line 231-289: Make sure that references follows the journal guidelines. There looks inconsistency currently.
The mistakes in References was corrected
Beside above other issues are:
If it is absolute power, what does negative value means, please mention it in your results. You may also discuss the relative power of EEG signal with respect electrode position.
The absolute power values are always positives, the difference that results from the comparison of the basal values against the results after the acute and subchronic intake of dark chocolate can adopt negative or positive values depending on whether there is an increase or decrease of the absolute power. On the other hand, although the relative power was obtained, we prefer to use the absolute power since contrary to what happens with the relative power, it is independent of the values of the other frequency bands.
Control is very important for this type of experiment. Generally (n=3 or 4) will also be acceptable for this purpose, please provide it.
The analysis of the cerebral oscillations of a control group (n = 5 ) was carried out and added in the corresponding sections.
Sincerely,
Efraín Santiago-Rodríguez, PhD
Diagnóstico, Tratamiento e Investigación Neurológica, A.C.
Prolongación Pino Suárez No. 357-A Col. Galindas
Querétaro, México, C.P. 76230
Tel.: +52-442-2426567
E-mail: efmx2000@gmail.com
Reviewer 2 Report
Please, provide the paper with EEG raw data.
Author Response
October 29th, 2018
Ms. Christine Fu
Assistant Editor, Foods
Dear Ms Christine Fu:
In response to the comments of the reviewers regarding our manuscript ID: Foods-380411 entitled “Effects of dark chocolate intake on brain electrical oscillations in healthy people”, we have made the following corrections to the manuscript (see below). We have included the revised version of the manuscript.
Reviewer 2
Please, provide the paper with EEG raw data.
EEG raw data is provided in attached file.
Sincerely,
Efraín Santiago-Rodríguez, PhD
Diagnóstico, Tratamiento e Investigación Neurológica, A.C.
Prolongación Pino Suárez No. 357-A Col. Galindas
Querétaro, México, C.P. 76230
Tel.: +52-442-2426567
E-mail: efmx2000@gmail.com

Round 2
Reviewer 1 Report
Dear authors:
I would like to thank you for addressing most of my concerns.
Few minor concerns I have now:
Line 102: It looks like you have done statistic at 5% level of significance and at other levels as of significance. Please mention if you have done test at different significance level.
Line 143-144: The wording "Values t", "Alpha y Beta", "intake. * t". These wording should be more clear. You may also replace " (p = 0.05). (n = 20) (ANOVA)" with (p = 0.05, n=20, ANOVA-test).
Line 108, 251: type error for "grup". Please check if similar errors has occurred throughout text.
Author Response
Dear authors:
I would like to thank you for addressing most of my concerns.
Few minor concerns I have now:
Line 102: It looks like you have done statistic at 5% level of significance and at other levels as of significance. Please mention if you have done test at different significance level.
The significance level of 0.05 was used. By error in the control group a level of significance of p> 0.10 was written. This error was corrected in the summary section, Material and Methods and in the corresponding figures.
Line 143-144: The wording "Values t" , "Alpha y Beta", "intake. * t". These wording should be more clear. You may also replace " (p = 0.05). (n = 20) (ANOVA)" with (p = 0.05, n=20, ANOVA-test).
The sentence was rewritten.
Line 108, 251: type error for "grup". Please check if similar errors has occurred throughout text.
The errors were corrected. The entire text was reviewed and errors of the same type were also corrected.